**Subject Category:**
Biology (whole organism)

plant science

*Ginkgo biloba* L., ginkgo nut, amino acid, protein quality, cultivar

**Author for correspondence:**
Li Xu
e-mail: xuliqby@163.com

# Protein content and amino acids profile in 10 cultivars of ginkgo (*Ginkgo biloba* L.) nut from China

Mengyi Zhou[1,2], Tongtong Hua[3], Xiaofang Ma[2], Haijun Sun[2] and Li Xu[2]

[1]College of Chemical Engineering, [2]Advanced Analysis and Testing Center, and [3]College of Forestry, Nanjing Forestry University, Nanjing, Jiangsu 210037, People's Republic of China

(iD) MZ, 0000-0001-9367-0350

As a traditional food and medicine source, ginkgo (*Ginkgo biloba* L.) nut is popularly consumed in East Asia. The aim of this work is to characterize protein content and amino acids profile in 10 ginkgo nut cultivars, named successively as no. 1 to no. 10. There were observed differences among the cultivars with respect to the contents of protein and amino acids, except Cys. The no. 6 cultivar presented the highest protein content (22.1 g/100 g DW), while the no. 9 had the lowest (16.2 g/100 g DW). The contents of EAA and NEAA were revealed to vary in the range of 14.3–26.2 and 21.4–41.1 g/100 g protein, respectively. The most abundant EAA was Leu, and the first limiting amino acid was Lys. The level of Arg was attractive, especially in the no. 5 cultivar (1741 mg/100 g DW) where it is comparable to hazelnut and pistachio. As confirmed by AAS and EAAI, the no. 5 cultivar presented the best amino acids profile and protein quality among these cultivars. These results have relevance to the scientific development and application of ginkgo nuts in the food industry.

## 1. Introduction

*Ginkgo biloba* L., which belongs to the botanical family of Ginkgoceae, has flourished on this planet for over 150 million years, and is perceived as a 'living fossil' [1]. The tree originates in China, and has been cultivated throughout Korea, Japan, Europe and the USA. Despite the medicinal value of the ginkgo tree being described in the Chinese Materia Medica more than 2000 years ago, its value has only been embraced by modern medicine in the last couple of decades [2]. Nowadays, it is among the best-selling herbal medications in the world [3]. Each

**Table 1.** Description of the ginkgo cultivars.

| number | cultivar name | mother tree origin |
|---|---|---|
| 1 | Taixing-4 | Jiangsu Agricultural College, Taixing County, Jiangsu Province |
| 2 | Dongtinghuang-2 | Jiangsu Agricultural College, Taixing County, Jiangsu Province |
| 3 | Yan'an | Yan'an Farm Board, Yan'an City, Shaanxi Province |
| 4 | Jiangzhong-1 | Yan'an Farm Board, Yan'an City, Shaanxi Province |
| 5 | Cao-2 | Caolou, Pizhou County, Jiangsu Province |
| 6 | Tancheng-300 | Huayuan, Tancheng County, Shandong Province |
| 7 | Yezi Yinxing | Yiyuan County, Shandong Province |
| 8 | Xincun-18 | Xincun, Tancheng County, Shandong Province |
| 9 | Changxing-1 | Changxing Forestry Bureau, Changxing County, Zhejiang Province |
| 10 | Changxing-5 | Changxing Forestry Bureau, Changxing County, Zhejiang Province |

part of the ginkgo tree has a particular medicinal property, especially its leaf and nut. To date, most of the available reports on the ginkgo tree have focused on its leaf extract, and its beneficial effects on neurodegenerative diseases, cardiovascular diseases, mood and memory have been confirmed [4–7]. The ginkgo nut has also shown promise for the prevention of cardiovascular and neurological disorders, although scientific research is limited [8,9].

The ginkgo nut has an orange flesh surrounding a hard tan shell, and its annual global yield is over 14 kt, more than 90% of which is produced in China [10]. As a traditional Chinese medicinal material, the ginkgo nut has been used for pulmonary diseases such as asthma and coughs, which is mentioned in the Compendium of Materia Medica [11,12]. Based on an ancient Chinese saying that 'Medicine and food are isogenic', the ginkgo nut is popularly consumed after fermentation or cooking in China and its neighbouring countries such as Japan and Korea [13]. Owing to its particular properties, the interest in the ginkgo nut has risen in Western medicine.

Nutrition consumed from foods plays a pivotal role for fundamental vital functions. The nutritional value of a food is mainly associated with its protein content and amino acid composition, which are essential components involved in promoting optimal health [14]. Knowing the amino acid profile as well as the protein quality of a food is a key criterion for assessing its nutritional value, and further facilitates policy formulation in the nutrition, food and health-related areas [15,16]. It has been recommended by FAO that the data on digestible or bioavailable amino acids be listed in food tables [17]. Although the ginkgo nut is not a protein-rich vegetable [18], considering its special status, the available information in terms of its amino acids and protein is insufficient. The primary aim of this work was to evaluate the protein content and amino acids profile of the ginkgo nut in order to supplement existing information. Taking into account the cultivar diversity, 10 common cultivars of ginkgo cultivated under the same field conditions were selected in this paper.

## 2. Material and methods

### 2.1. Samples and sample preparation

A total of 10 ginkgo cultivars were randomly selected from among those cultivated on Pizhou Ginkgo biloba Seedling Base, which are cultivated at a plantation site under the same agronomic conditions (table 1). The samples used in this paper were collected in October 2016. Approximately 80 fruits were harvested randomly from three plants of each cultivar. After the harvest, the fresh ginkgo fruits were transported to the laboratory, where the edible ginkgo nuts were shelled manually, dried by freeze dryer and stored at 4°C. Three portions were randomly selected from these for the following analysis.

### 2.2. Reagents

Ultrapure water used for the preparation of all solutions were obtained from a Milli-Q purifier (Millipore, Eschborn, Germany). Eluent buffer A (0.12 M citric acid−sodium citrate buffer at pH 3.45) and B (0.20 M

citric acid–sodium citrate buffer at pH 10.85), regeneration buffer D (0.50 M sodium hydroxide), sample dilution buffer (0.12 M citric acid–sodium citrate buffer at pH 2.20) and ninhydrin (triketohydrindene hydrate) were purchased from Sykam GmbH Company (Germany). Unless otherwise indicated, reagents used in this work were of analytical reagent grade quality.

A working standard solution was diluted from the Amino Acid Standard Hydrolysate (Sykam, Germany), which contains 2.5 mM of each amino acid including aspartic acid (Asp), threonine (Thr), serine (Ser), glutamic acid (Glu), proline (Pro), glycine (Gly), alanine (Ala), cysteine (Cys), valine (Val), methionine (Met), isoleucine (Ile), leucine (Leu), tyrosine (Tyr), phenylalanine (Phe), histidine (His), lysine (Lys) and arginine (Arg).

## 2.3. Analysis

### 2.3.1. Crude protein

As described by Wong & Cheung [19], the crude protein (CP) content was determined by multiplying the nitrogen content by a conversion factor of 6.25, while the nitrogen content was detected by a Perkin-Elmer 2400 automatic element analyser (Perkin-Elmer, CT, USA). The CP content in ginkgo nut was reported on a dry weight (DW) basis.

### 2.3.2. Amino acids composition

Freeze-dried and milled samples were used to analyse amino acids composition, according to the procedure reported by Galdón *et al*. [20] with slight modifications. A sample (50 mg) was weighed in a Pyrex glass tube with a Teflon-lined screw cap, followed by the addition of 5 ml of 6 M HCl. The tube was frozen by liquid nitrogen and sealed under vacuum, and then placed in a drying oven at 110°C for 24 h. Each hydrolysed extract was filled up with ultrapure water to yield 10 ml, and filtered through a 0.45 mm filter. An aliquot of 0.5 ml was transferred to a fresh 1.5 ml EP tube to evaporate in a vacuum evaporator, and the dried residue obtained was dissolved in 5 ml sample dilution buffer. Finally, 50 µl filtered sample or standard solution was loaded into a Sykam S433D amino acid analyser (Sykam, Germany).

The concentrations of amino acids were measured by ion exchange chromatography, followed by post-column ninhydrin derivation. Amino acids in samples were separated by a $4.6 \times 150$ mm cation separation column LCA K06/Na (Sykam) using a ternary gradient of eluent A, eluent B and buffer D at $0.45$ ml min$^{-1}$ flow rate as follows: 0–11 min, 0% B; 11–17 min, 15% B; l7–23 min, 20% B; 23–27 min, 33% B; 27–30 min, 80% B; 30–42 min, 100% B; 42–45 min, 100% D. The column temperature was raised from 57 to 74°C at 27 min. The separated amino acids were detected by a spectrophotometric detection at 440 nm (for Pro) or 570 nm (for the others). The concentrations of amino acids were expressed as g/100 g protein. Because of its degradation during acidic hydrolysis, no analysis concerning tryptophan was carried out.

## 2.4. Protein nutritional quality

Amino acid score (AAS) and essential amino acid index (EAAI) were calculated using the following equations, respectively [14]

$$\mathrm{AAS} = \frac{a_p}{a_s},$$

$$\mathrm{EAAI} = \sqrt[n]{\mathrm{Lys}_{1p}/\mathrm{Lys}_{1s} \times \mathrm{Tyr}_{2p}/\mathrm{Tyr}_{2s} \times \cdots \times \mathrm{His}_{np}/\mathrm{His}_{ns}} \times 100\%,$$

where $a$ is an essential amino acid (EAA), $p$ is the test protein, $s$ is the reference protein and $n$ is the number of amino acids entering into the calculation. The reference protein used was the FAO/WHO amino acid scoring pattern from the 2007 WHO/FAO/UNU report [21].

## 2.5. Statistical analysis

All data were reported as the means with standard errors of the mean (s.e.m.). The statistical significance among samples was tested at the 0.05 probability level by one-way analysis of variance (ANOVA) using the *Tukey* test (SPSS v. 21). The data obtained from the analysis of the amino acid concentrations were furtherly analysed statistically by hierarchical cluster analysis using Ward's method and squared

Euclidean distance. The dendrogram similarity scale ranges from zero (greater similarity) to 25 (lower similarity) by SPSS.

# 3. Results and discussion

According to the available records, the history of *Ginkgo biloba* L. cultivation reaches back over 2000 years. In China, a series of growing centres have been developed, including in Taixing and Pizhou counties, Jiangsu province; Tancheng county, Shandong province; Zhuji and Changxing counties, Zhejiang province; Shaanxi province, etc. [22]. The cultivars analysed in this paper were originally grown in these centres (table 1).

## 3.1. Crude protein analysis

It is well known that protein, one of most vital macronutrients from food for humans, is involved in various physiological functions. As shown in table 2, the content of CP ranged from 16.2 up to 22.1 g/100 g of edible portion on a DW basis. Among these cultivars, the no. 6 and no. 5 ranked first and second in terms of the CP content, while the no. 9 ranked last. The CP content of ginkgo nut, on average 19.7% of dry matter, was higher than that reported by Deng *et al.* [18], who found that the protein content was $11.6 \pm 0.26\%$ in defatted ginkgo flour. This might be due to the differences in analytical procedure and data expression.

## 3.2. Amino acid concentrations

### 3.2.1. General

The amino acids analysed here are divided into two categories as reported by Shaheen *et al.* [15]: Cys, His, Ile, Leu, Lys, Met, Phe, Thr, Tyr and Val as EAAs, and Ala, Arg, Asp, Glu, Gly, Pro and Ser as non-essential amino acids (NEAAs).

The amino acids were measured through chromatography separation coupled with post-column derivatization. Figure 1 illustrates a chromatogram corresponding to a mixture of the 17 amino acid standards (A) and a ginkgo nut sample (B), where a good resolution and separation of all the identified amino acids in the ginkgo nut sample determined by the procedure described in §2.3.2 could be observed.

The protein quality, which represents to some extent the nutritive value of a food, is governed by parameters including amino acid composition, digestibility, source and processing [14]. Among these, amino acid composition is essential to the assessment of protein quality on account of its relation to other food properties such as digestibility [14]. Table 2 reveals the amino acids profile of ginkgo nuts of the 10 cultivars. The results of one-way ANOVA comparing the mean concentrations between the cultivars are also concluded. In general, distinct variations in mean values of the concentrations for all the amino acids except Cys were observed between the cultivars analysed. Taking into consideration that the amino acid composition of food protein could vary not only with genetic background but also environmental growth conditions such as soil type [23], ginkgo nuts analysed in this study were harvested in the same plantation at the same time. Thus, a conclusion might be drawn that the genetic characteristics of the ginkgo cultivars decisively affect the amino acids profile.

### 3.2.2. Essential amino acids

In contrast with plants, both animals and humans require certain quantities of EAAs through dietary protein consumption. Thus, the quality of a dietary protein is to some extent determined by the content of EAAs in that protein. Table 1 shows that the sum of EAAs ranged from 14.3 in the no. 10 to 26.2 g/100 g protein in the no. 5. The no. 5 stood out among the analysed cultivars due to the fact that it had considerably high concentrations in the following amino acids: Leu, Val, Ile, Pre, Lys, Tyr and His, whereas Thr, Cys and Met were of intermediate level. Of all the individual EAAs, Leu, which was involved in the protein synthesis [24], was the most highly available EAA in the ginkgo nut. The results were as expected in accordance with the available literature for several edible nut seeds [25]. Sulfur amino acids (SAAs), Met and Cys, are a critical group of amino acids, which could be easily lost from the body. SAAs are generally insufficient in plant source foods [26], and ginkgo nut was no exception in that the participation of Met and Cys in the total amino acids ranged

**Table 2.** The contents (mean ± s.e.m., $n = 3$) of CP and amino acids according to the cultivars. All data are expressed as g/100 g protein, except CP which is expressed as g/100 g DW. Results in the same row with the same superscript were not statistically significant ($p < 0.05$) according to the classification obtained by the Tukey test. A, aromatic amino acids: Phe + Tyr; B, sulfur amino acids: Cys + Met; C, essential amino acids: Thr + Cys + Val + Met + Ile + Leu + Tyr + Phe + His + Lys; D, non-essential amino acids: Asp + Ser + Glu + Pro + Gly + Ala + Arg.

| items | cultivars | | | | | | | | | |
|---|---|---|---|---|---|---|---|---|---|---|
| | 1 | 2 | 3 | 4 | 5 | 6 | 7 | 8 | 9 | 10 |
| CP | 18.5 ± 1.17[ab] | 20.4 ± 0.51[ab] | 20.5 ± 0.43[ab] | 19.5 ± 0.90[ab] | 21.9 ± 1.90[b] | 22.1 ± 0.35[b] | 18.5 ± 1.10[ab] | 21.1 ± 1.08[ab] | 16.2 ± 1.53[a] | 18.2 ± 1.23[ab] |
| AA | | | | | | | | | | |
| Asp | 6.42 ± 0.21[bc] | 8.32 ± 0.07[de] | 7.27 ± 0.15[cd] | 7.37 ± 0.04[cd] | 9.71 ± 0.27[e] | 6.79 ± 0.23[cd] | 6.41 ± 0.24[bc] | 5.93 ± 0.24[bc] | 4.91 ± 0.17[ab] | 4.05 ± 0.06[a] |
| Thr | 2.29 ± 0.07[ab] | 3.06 ± 0.18[b] | 2.25 ± 0.07[ab] | 3.07 ± 0.07[b] | 2.76 ± 0.07[ab] | 2.56 ± 0.09[ab] | 2.20 ± 0.05[ab] | 2.86 ± 0.12[b] | 2.43 ± 0.10[ab] | 1.92 ± 0.04[a] |
| Ser | 1.51 ± 0.05[ab] | 2.44 ± 0.29[abca] | 1.44 ± 0.06[ab] | 2.67 ± 0.08[bc] | 1.59 ± 0.06[abc] | 1.93 ± 0.16[abc] | 1.37 ± 0.07[a] | 2.83 ± 0.13[c] | 2.46 ± 0.09[abc] | 2.02 ± 0.04[abc] |
| Glu | 7.67 ± 0.29[ab] | 10.1 ± 0.23[bcd] | 8.61 ± 0.18[bc] | 10.4 ± 0.17[cd] | 11.3 ± 0.19[e] | 9.23 ± 0.07[bcd] | 8.16 ± 0.15[abc] | 8.92 ± 0.48[bcd] | 5.86 ± 0.27[a] | 6.02 ± 0.11[a] |
| Pro | 2.61 ± 0.11[ab] | 3.11 ± 0.19[b] | 3.10 ± 0.09[b] | 2.89 ± 0.17[ab] | 3.20 ± 0.15[b] | 2.52 ± 0.07[ab] | 2.31 ± 0.10[ab] | 2.63 ± 0.06[ab] | 1.96 ± 0.16[ab] | 1.79 ± 0.11[a] |
| Gly | 2.04 ± 0.07[ab] | 3.02 ± 0.10[cd] | 2.34 ± 0.05[abcd] | 2.85 ± 0.05[bcd] | 3.26 ± 0.07[d] | 2.33 ± 0.07[abc] | 2.00 ± 0.07[ab] | 2.14 ± 0.18[abc] | 1.71 ± 0.13[a] | 1.74 ± 0.02[a] |
| Ala | 2.91 ± 0.10[abc] | 3.65 ± 0.03[cd] | 3.19 ± 0.09[bcd] | 3.72 ± 0.12[cd] | 4.08 ± 0.08[d] | 3.42 ± 0.05[cd] | 3.05 ± 0.03[bc] | 3.51 ± 0.16[cd] | 2.50 ± 0.11[ab] | 2.00 ± 0.02[a] |
| Cys | 0.16 ± 0.01[a] | 0.10 ± 0.01[a] | 0.14 ± 0.01[a] | 0.09 ± 0.01[a] | 0.10 ± 0.00[a] | 0.09 ± 0.01[a] | 0.12 ± 0.01[a] | 0.16 ± 0.01[a] | 0.17 ± 0.01[a] | 0.18 ± 0.02[a] |
| Val | 2.95 ± 0.09[ab] | 4.00 ± 0.06[cd] | 3.61 ± 0.11[bc] | 3.31 ± 0.09[abc] | 4.78 ± 0.09[d] | 3.71 ± 0.13[bc] | 3.23 ± 0.11[abc] | 3.80 ± 0.11[bc] | 3.34 ± 0.08[abc] | 2.55 ± 0.02[a] |
| Met | 0.75 ± 0.01[abc] | 0.96 ± 0.03[bc] | 0.62 ± 0.05[ab] | 1.04 ± 0.01[c] | 0.96 ± 0.08[bc] | 0.74 ± 0.05[abc] | 0.75 ± 0.04[abc] | 0.76 ± 0.05[abc] | 0.60 ± 0.04[ab] | 0.44 ± 0.01[a] |
| Ile | 1.95 ± 0.07[abc] | 2.82 ± 0.03[de] | 2.54 ± 0.08[cd] | 2.53 ± 0.06[cd] | 3.32 ± 0.07[e] | 2.54 ± 0.05[cd] | 2.25 ± 0.08[bcd] | 2.69 ± 0.09[de] | 1.70 ± 0.11[ab] | 1.53 ± 0.01[a] |
| Leu | 3.63 ± 0.12[abc] | 4.96 ± 0.05[de] | 4.16 ± 0.09[cd] | 4.78 ± 0.12[de] | 5.56 ± 0.13[e] | 4.39 ± 0.00[cd] | 3.87 ± 0.10[bcd] | 4.51 ± 0.19[cde] | 2.95 ± 0.18[ab] | 2.59 ± 0.02[a] |
| Tyr | 1.32 ± 0.02[abc] | 1.73 ± 0.04[cde] | 1.49 ± 0.03[bcd] | 1.84 ± 0.04[de] | 2.05 ± 0.06[e] | 1.50 ± 0.01[cd] | 1.43 ± 0.02[bcd] | 1.47 ± 0.08[bcd] | 1.05 ± 0.06[ab] | 0.91 ± 0.04[a] |
| Phe | 1.80 ± 0.05[abc] | 2.39 ± 0.03[cd] | 2.04 ± 0.06[bcd] | 2.20 ± 0.04[bcd] | 2.59 ± 0.06[d] | 1.89 ± 0.01[abc] | 1.84 ± 0.03[abc] | 1.94 ± 0.11[abc] | 1.62 ± 0.10[ab] | 1.52 ± 0.01[a] |
| His | 1.51 ± 0.01[ab] | 1.88 ± 0.02[ab] | 1.74 ± 0.04[ab] | 1.80 ± 0.06[ab] | 1.96 ± 0.05[b] | 1.61 ± 0.02[ab] | 1.52 ± 0.03[ab] | 1.68 ± 0.05[ab] | 1.61 ± 0.10[ab] | 1.40 ± 0.03[a] |
| Lys | 1.39 ± 0.07[ab] | 1.76 ± 0.09[ab] | 1.63 ± 0.08[ab] | 1.74 ± 0.06[ab] | 2.14 ± 0.07[b] | 1.49 ± 0.05[ab] | 1.40 ± 0.03[ab] | 1.57 ± 0.12[ab] | 1.40 ± 0.13[ab] | 1.29 ± 0.06[a] |
| Arg | 4.86 ± 0.26[ab] | 6.77 ± 0.23[bc] | 6.66 ± 0.26[bc] | 7.50 ± 0.12[c] | 7.94 ± 0.14[c] | 6.30 ± 0.02[bc] | 5.38 ± 0.21[ab] | 6.78 ± 0.35[bc] | 3.67 ± 0.17[a] | 3.77 ± 0.08[a] |
| ∑AAA[A] | 3.12 ± 0.14[bc] | 4.13 ± 0.13[cd] | 3.53 ± 0.19[bc] | 4.04 ± 0.15[cd] | 4.63 ± 0.23[d] | 3.39 ± 0.03[abc] | 3.27 ± 0.09[abc] | 3.41 ± 0.40[abc] | 2.67 ± 0.32[ab] | 2.43 ± 0.08[a] |
| ∑SAA[B] | 0.91 ± 0.02[ab] | 1.07 ± 0.07[b] | 0.76 ± 0.07[ab] | 1.12 ± 0.05[b] | 1.06 ± 0.16[b] | 0.83 ± 0.08[ab] | 0.88 ± 0.05[ab] | 0.92 ± 0.11[ab] | 0.78 ± 0.09[ab] | 0.61 ± 0.02[a] |
| ∑EAA[C] | 17.8 ± 0.99[abc] | 23.7 ± 0.64[de] | 20.2 ± 0.99[bcd] | 22.4 ± 0.90[cde] | 26.2 ± 1.29[e] | 20.5 ± 0.21[bcd] | 18.6 ± 0.79[abcd] | 21.5 ± 1.85[bcde] | 16.9 ± 1.31[ab] | 14.3 ± 0.25[a] |
| ∑NEAA[D] | 28.0 ± 2.06[ab] | 37.4 ± 1.89[cd] | 32.6 ± 1.56[bcd] | 37.4 ± 1.33[cd] | 41.1 ± 1.93[d] | 32.5 ± 0.33[bcd] | 28.7 ± 1.30[abc] | 32.8 ± 3.16[bcd] | 23.1 ± 1.80[a] | 21.4 ± 0.66[a] |
| ∑AA | 45.8 ± 3.04[abc] | 61.1 ± 2.53[de] | 52.8 ± 2.49[bcd] | 59.8 ± 2.23[de] | 67.3 ± 3.21[e] | 53.0 ± 0.14[bcd] | 47.3 ± 2.08[abcd] | 54.2 ± 5.01[cde] | 40.0 ± 2.92[ab] | 35.7 ± 0.81[ab] |

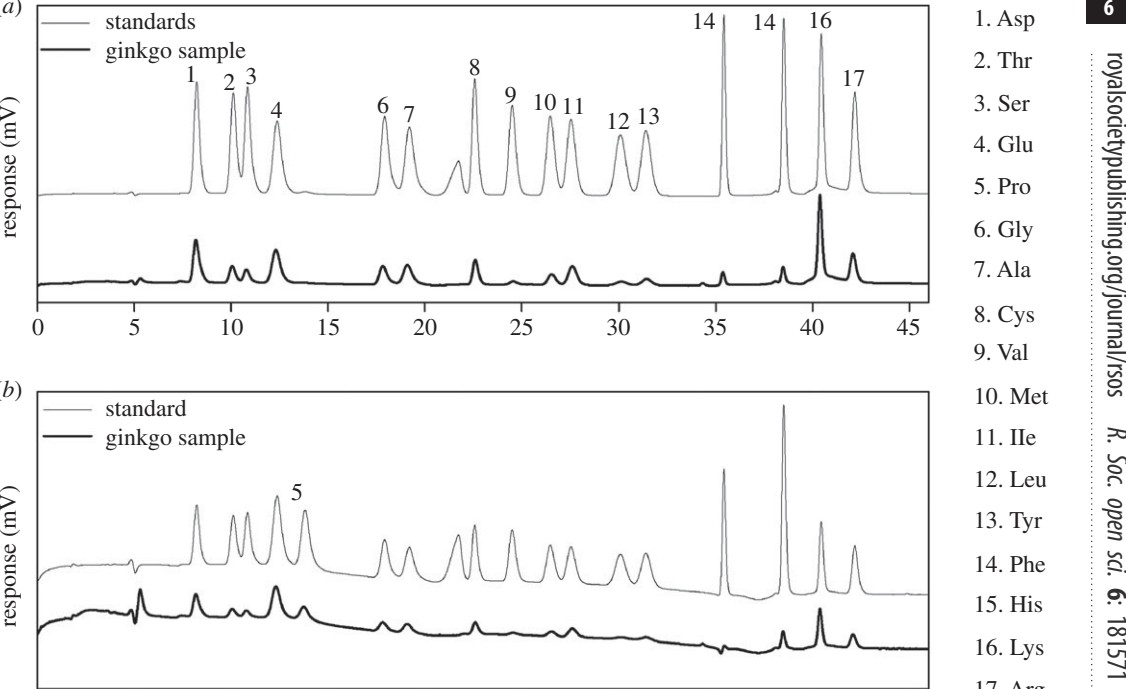

**Figure 1.** Liquid chromatograms of the amino acid standards (fine line) and a ginkgo sample (thick line) at a detector wavelength of 570 nm (*a*) or 440 nm (*b*).

between 1.17 and 1.64% and between 0.15 and 0.49%, respectively. Additionally, no significant difference in the proportion of EAAs in total amino acids was found between the cultivars.

### 3.2.3. Non-essential amino acids

NEAAs are those amino acids which can be synthesized in animals and humans. Emerging evidence indicates that NEAAs are essential not to the taste of foods but to the growth and health of animals and humans [27]. Again, marked differences in NEAAs content were found between the cultivars, e.g. between the no. 5 and no. 10, which were similar to that of EAAs (table 1). It is clear that among the cultivars, the no. 5 contained the highest sum of NEAAs, which could be attributed to its appreciable amounts of almost all the NEAAs except Ser. Ginkgo nut was found to be rich in Glu, a constituent of umami taste, which was obtained from the brown seaweed Kombu [28]. Another acidic amino acid Asp was ranked second in terms of its concentration, followed by that of Glu. The proportions of Glu and Asp ranged between 14.7 and 17.4% and between 10.9% and 14.4% of the total amino acids, respectively. Arg in ginkgo nut ranged from 3.67 g/100 g of protein in the no. 9 to 7.94 g/100 g of protein in the no. 5. Noteworthily, the no. 5 presented a rich source of Arg, 1741 mg of Arg/100 g of edible portion, which is comparable to hazelnut (1761 mg/100 g), pistachio (1812 mg/100 g) and whiting fish (2150 mg/182.6 g) [29]. Arg is a precursor of nitric oxide (NO), and NO is a conspicuous molecule due to its diverse bioactivities including immunological regulation and antioxidative. There is evidence showing that the health-related benefits of protein and amino acids from dietary foods are derived from their modulation of NO production [30]. Thus, the high Arg content of ginkgo nut, especially the no. 5 cultivar, offers opportunities for its application as part of a healthy, nutritionally-balanced diet.

### 3.2.4. Hierarchical cluster analysis

Hierarchical cluster analysis is quite a useful method for observing and analysing a collected data matrix, and can generate a hierarchy based on a prescribed set of steps and algorithms [31]. Ward algorithms with squared Euclidean distances is the best known and most often used cluster method for producing partition hierarchies [32]. This analysis was applied to group the ginkgo cultivars through

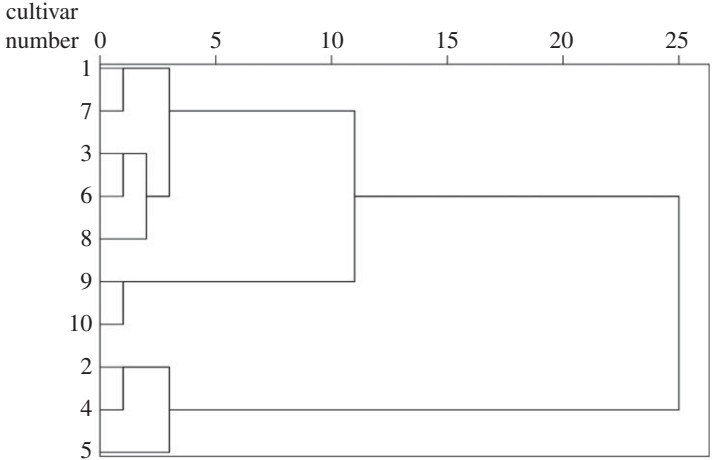

**Figure 2.** Dendrogram of hierarchical clustering results of the 10 ginkgo nut cultivars, using Ward's method and squared Euclidean distance by SPSS software.

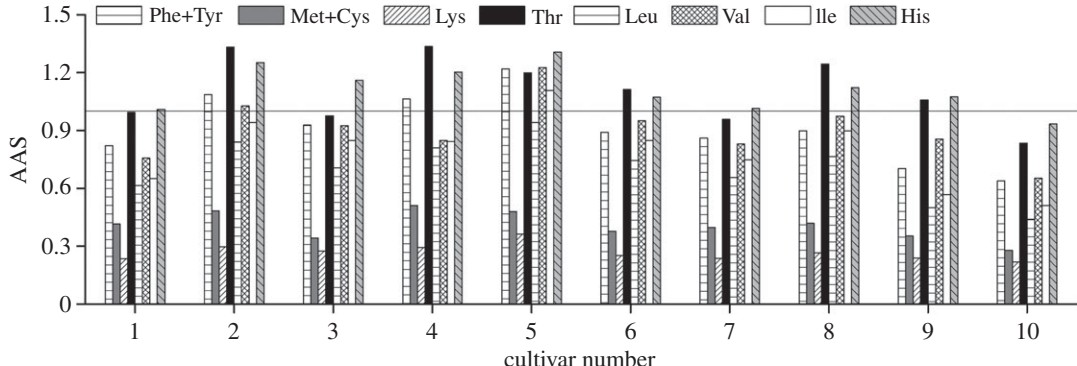

**Figure 3.** AAS of the ginkgo cultivars studied. WHO/FAO/UNO adult maintenance pattern are expressed as g amino acid/100 g protein: Phe + Tyr, 3.8; Met + Cys, 2.7; Lys, 4.5; Thr, 2.3; Leu, 5.9; Val, 3.9; Ile, 3.0; His, 1.5 (WHO/FAO, 2007 [21]).

their similarities on the amino acid concentration. As shown in figure 2, there were three clusters with an above, larger one divided into two subclusters and a smaller one. The cluster which consisted of the five cultivars of the no. 1, no. 3, no. 6, no. 7 and no. 8 represented the middle level of EAAs and total amino acids. The no. 9 and no. 10, which contained the lowest level of EAAs and total amino acids among these cultivars, grouped together and formed a cluster. The no. 5 clustered into the group with the no. 3 and no. 6, which indicated that among these cultivars, they had a great similarity in terms of their abundance of EAAs and total amino acids. The hierarchical cluster analysis results visually exhibited the degree of similarity between the analysed ginkgo cultivars.

## 3.3. Nutritive quality of ginkgo nut protein

### 3.3.1. Amino acid score and limiting amino acid

As a chemical index, AAS is applied to simply measure the quality of protein in human nutrition based on comparison of all the EAAs with respect to the reference protein of WHO/FAO/UNU [21]. AAS could also point at the limiting amino acid. Figure 3 provides a comparative overview of AAS values of individual EAA for each cultivar. It was conspicuous that the AAS for more than half of EAAs in the no. 5 exceeded 100%. His, which participate in various physiological activities including protein interactions as well as tissue growth and repair [33], was calculated to be higher than that suggested by WHO/FAO/UNO in all cultivars but the no. 10. Lys was the first limiting amino acid for ginkgo nut, which is consistent with the published report for cashew nut, hazelnut, pine nut and walnut [29].

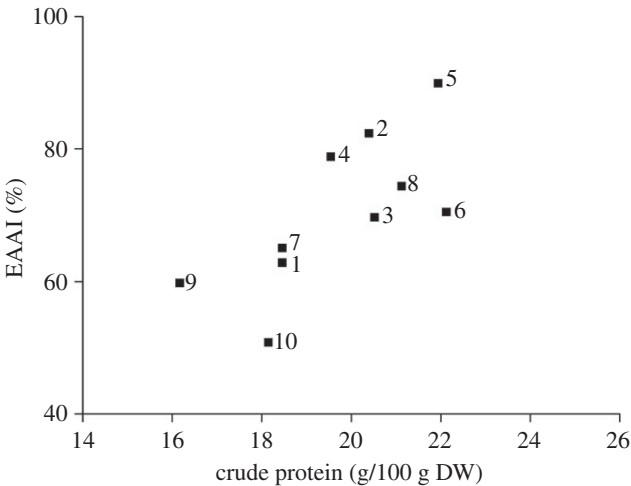

**Figure 4.** EAAI of the protein in gingko nuts of different varieties in relation to CP content.

The AAS for Lys varied between 21.9% for the no. 10 and 36.3% for the no. 5. Thus, the proteins from the no. 5 cultivar presented the best amino acids profile.

### 3.3.2. Essential amino acid index

With respect to protein quality, EAAI, proposed by Oser [34], focuses on the content of all the EAAs in proteins as an integrated whole for their nutritional assessment. A high quality and efficiency of protein is generally characterized by a high value of EAAI [35]. A simple scoring scheme is stated by Brown *et al.* [36]: high quality when a protein has an EAAI value of greater than 0.90, moderate nutritional value when 0.70–0.89 and low quality when less than 0.70. The EAAI of the no. 2 (0.82), no. 4 (0.79), no. 6 (0.70) and no. 8 (0.74) exceeded those of the rest of the studied cultivars except the no. 5 (figure 4). The EAAI value of the no. 5 was 0.90, which was higher than that for rice (0.83), and was compared favourably with cow's milk (0.89) [37].

Also, figure 4 exhibits the relationship between the EAAI and protein content in ginkgo nut whereby EAAI exerted a slightly increase trend with increasing protein content in the nuts of studied cultivars. Among these cultivars, the no. 5 clearly exhibited a relatively high EAAI value and high protein content.

## 4. Conclusion

In this paper, the CP content and amino acids profile of ginkgo nut were analysed, and evident differences between the 10 selected cultivars on these two aspects were observed. Leu was the most available EAA in ginkgo nut, while Lys was the first limiting amino acid. Among NEAAs, the concentration of Arg was satisfactory in ginkgo nut, especially the no. 5 cultivar. With respect to the quality of protein, AAS and EAAI were calculated based on the reference protein suggested by WHO/FAO/UNO 2007. On the basis of the values of AAS and EAAI, the no. 5 cultivar was nutritionally superior to other ginkgo nut cultivars. To our knowledge, the data on the protein and amino acids in ginkgo nut are seldom found in the published literature, and there was no report for the comparative analysis of these 10 cultivars. Hence, the information given in this paper is of reference value with a view to scientific development and application of ginkgo nuts in the food industry.

Data accessibility. The raw values of N content and amino acid concentration for each sample of ginkgo nuts are available in the electronic supplementary material. All other relevant data are disclosed within the paper.

Authors' contributions. M.Z. and L.X. conceived and designed the research; M.Z., T.H., X.M. and H.S. ran the experiment and carried out the data analysis; M.Z. and L.X. wrote and revised the manuscript; L.X. had primary responsibility for the final content. All authors read and approved the final version of the manuscript.

Competing interests. We have no competing interests.

Funding. This work was kindly supported by research projects of Higher School Natural Science of Jiangsu Province (15KJB530009) and Youth Science and Technology Innovation Fund of Nanjing Forestry University (CX2016013).

Acknowledgements. We acknowledge Advanced Analysis and Testing Center of Nanjing Forestry University for the required research facilities. And, we thank the anonymous reviewers for valuable suggestions.

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
