## [Reviewer comments · Royal Society Open Science]

Review History

RSOS-181571.R0 (Original submission)

Review form: Reviewer 1

Is the manuscript scientifically sound in its present form?

Yes

Are the interpretations and conclusions justified by the results?

Yes

Is the language acceptable?

Yes

Is it clear how to access all supporting data?

Yes

Do you have any ethical concerns with this paper?

No

Have you any concerns about statistical analyses in this paper?

I do not feel qualified to assess the statistics

Recommendation?

Accept with minor revision (please list in comments)

Comments to the Author(s)

The article (Manuscript ID RSOS-181571) on "Protein content and amino acids profile in ten cultivars of ginkgo (*Ginkgo biloba* L.) nut from China" is relevant to publication in Royal Society Open Science after minor correction.

Submitted for review work on "Protein content and amino acids profile in ten cultivars of ginkgo (*Ginkgo biloba* L.) nut from China" in an interesting and substantively as well as statistically correct way describes the amino acid composition of ginkgo biloba nuts, their nutritional value and compares the content of individual amino acids as well as amino acid groups in the amino acid profile of nuts of 10 tested ginkgo biloba varieties. The research material, the methodology of the conducted research and the presentation of the results do not raise objections of the reviewer. However, in the Introduction part of the article there is a lack of broader information regarding the research conducted so far by other authors in the analysis of the amino acid composition of ginkgo biloba nuts and their nutritional value. Also, some more details in a discussion should be added. The work has a high cognitive value.

Detailed comments:

Title: Probably a sentence of "Insert the title of your article here" is not necessary?

Page 2, line 44: "amino acids" is repeated

Page 3, line 2: there is a mistake in a word "degradation"

Results and Discussion

Page 4 line 36: what was CP content reported by Deng [18]? It needs better information.

Page 4 lines 7-8: a discussion needs deeper explanation

Page 4, line 5: there is "Pre" but shall be "Pro"

There are errors in the spelling of the word "table":

Page 7, lines 2, 18

Page 8, line 2

Review form: Reviewer 2**Is the manuscript scientifically sound in its present form?**

Yes

Are the interpretations and conclusions justified by the results?

Yes

Is the language acceptable?

Yes

Is it clear how to access all supporting data?

Not Applicable

Do you have any ethical concerns with this paper?

No

Have you any concerns about statistical analyses in this paper?

No

Recommendation?

Accept with minor revision (please list in comments)

Comments to the Author(s)

This manuscript describes protein content and amino acids profile in ten cultivars of ginkgo (*Ginkgo biloba* L.) nut from China. The manuscript will be acceptable after minor revision.

Page 1, Summary and throughout the whole manuscript, including Table 2, all the analytical data should be given the three significant figures, e.g. 22.13 should be 22.1.

Page 2, section 3.1. Samples and sample preparation

How many fruits and/or what amounts of fruits were sampled from how many plants of each cultivar should be given.

Page 3, section 4. Results and Discussion

The authors have not given any explanation/discussion as to why there is difference in the amino acid and protein contents of different cultivars of the plant? It will be useful to correlate the variations in amino acids with soil parameters (composition), climatic conditions, rain fall, temperature, pH, fertilizers used and so on.

Decision letter (RSOS-181571.R0)

11-Feb-2019

Dear Dr Zhou

On behalf of the Editors, I am pleased to inform you that your Manuscript RSOS-181571 entitled "Protein content and amino acids profile in ten cultivars of ginkgo (*Ginkgo biloba* L.) nut from China" has been accepted for publication in Royal Society Open Science subject to minor revision in accordance with the referee suggestions. Please find the referees' comments at the end of this email.

The reviewers and handling editors have recommended publication, but also suggest some minor revisions to your manuscript. Therefore, I invite you to respond to the comments and revise your manuscript.

- Ethics statement

- Data accessibility

It is a condition of publication that all supporting data are made available either as supplementary information or preferably in a suitable permanent repository. The data accessibility section should state where the article's supporting data can be accessed. This section should also include details, where possible of where to access other relevant research materials such as statistical tools, protocols, software etc can be accessed. If the data has been deposited in an external repository this section should list the database, accession number and link to the DOI for all data from the article that has been made publicly available. Data sets that have been

deposited in an external repository and have a DOI should also be appropriately cited in the manuscript and included in the reference list.

If you wish to submit your supporting data or code to Dryad (<http://datadryad.org/>), or modify your current submission to dryad, please use the following link:
<http://datadryad.org/submit?journalID=RSOS&manu=RSOS-181571>

- **Competing interests**

- **Authors' contributions**

- **Acknowledgements**

- **Funding statement**

Because the schedule for publication is very tight, it is a condition of publication that you submit the revised version of your manuscript before 20-Feb-2019. Please note that the revision deadline will expire at 00.00am on this date. If you do not think you will be able to meet this date please let me know immediately.

When submitting your revised manuscript, you will be able to respond to the comments made by

the referees and upload a file "Response to Referees" in "Section 6 - File Upload". You can use this to document any changes you make to the original manuscript. In order to expedite the processing of the revised manuscript, please be as specific as possible in your response to the referees. We strongly recommend uploading two versions of your revised manuscript:

Kind regards,
Royal Society Open Science Editorial Office

on behalf of Professor Kevin Padian (Subject Editor)
openscience@royalsociety.org

Associate Editor Comments to Author:

Both reviewers find merit in publishing your paper, though they have a number of recommendations for improvement. Please ensure you respond to these queries - both by incorporating the requested changes and in your point-by-point response to the reviewers. If you do not include an change, you must fully address this via a scientific rebuttal in the response.

Reviewer comments to Author:

Reviewer: 1

Comments to the Author(s)

The article (Manuscript ID RSOS-181571) on "Protein content and amino acids profile in ten cultivars of ginkgo (*Ginkgo biloba* L.) nut from China" is relevant to publication in Royal Society Open Science after minor correction.

Submitted for review work on "Protein content and amino acids profile in ten cultivars of ginkgo (*Ginkgo biloba* L.) nut from China" in an interesting and substantively as well as statistically correct way describes the amino acid composition of ginkgo biloba nuts, their nutritional value and compares the content of individual amino acids as well as amino acid groups in the amino acid profile of nuts of 10 tested ginkgo biloba varieties. The research material, the methodology of the conducted research and the presentation of the results do not raise objections of the reviewer. However, in the Introduction part of the article there is a lack of broader information regarding the research conducted so far by other authors in the analysis of the amino acid composition of ginkgo biloba nuts and their nutritional value. Also, some more details in a discussion should be added. The work has a high cognitive value.

Detailed comments:

Title: Probably a sentence of "Insert the title of your article here" is not necessary?

Page 2, line 44: "amino acids" is repeated

Page 3, line 2: there is a mistake in a word "degradation"

Results and Discussion

Page 4 line 36: what was CP content reported by Deng [18]? It needs better information.

Page 4 lines 7-8: a discussion needs deeper explanation

Page 4, line 5: there is "Pre" but shall be "Pro"

There are errors in the spelling of the word "table":

Page 7, lines 2, 18

Page 8, line 2

Reviewer: 2

Comments to the Author(s)

This manuscript describes protein content and amino acids profile in ten cultivars of ginkgo (*Ginkgo biloba* L.) nut from China. The manuscript will be acceptable after minor revision.

Page 1, Summary and throughout the whole manuscript, including Table 2, all the analytical data should be given the three significant figures, e.g. 22.13 should be 22.1.

Page 2, section 3.1. Samples and sample preparation

How many fruits and/or what amounts of fruits were sampled from how many plants of each cultivar should be given.

Page 3, section 4. Results and Discussion

The authors have not given any explanation/discussion as to why there is difference in the amino acid and protein contents of different cultivars of the plant? It will be useful to correlate the variations in amino acids with soil parameters (composition), climatic conditions, rain fall, temperature, pH, fertilizers used and so on.

Author's Response to Decision Letter for (RSOS-181571.R0)

See Appendix A.

Decision letter (RSOS-181571.R1)

18-Feb-2019

Dear Dr Zhou,

I am pleased to inform you that your manuscript entitled "Protein content and amino acids profile in ten cultivars of ginkgo (*Ginkgo biloba* L.) nut from China" is now accepted for publication in Royal Society Open Science.

Kind regards,
Andrew Dunn
Senior Publishing Editor
Royal Society Open Science
openscience@royalsociety.org

on behalf of Prof Kevin Padian (Subject Editor)
openscience@royalsociety.org

Appendix A

We thank the editor and reviewers for their time and comments. The comments from reviewers are highly constructive, and we have acted on them to further improve the manuscript. The detailed responses to the concerns from them are provided as follows:

Reviewer 1

The article (Manuscript ID RSOS-181571) on “Protein content and amino acids profile in ten cultivars of ginkgo (*Ginkgo biloba* L.) nut from China” is relevant to publication in Royal Society Open Science after minor correction.

Submitted for review work on “Protein content and amino acids profile in ten cultivars of ginkgo (*Ginkgo biloba* L.) nut from China” in an interesting and substantively as well as statistically correct way describes the amino acid composition of ginkgo biloba nuts, their nutritional value and compares the content of individual amino acids as well as amino acid groups in the amino acid profile of nuts of 10 tested ginkgo biloba varieties. The research material, the methodology of the conducted research and the presentation of the results do not raise objections of the reviewer. However, in the Introduction part of the article there is a lack of broader information regarding the research conducted so far by other authors in the analysis of the amino acid composition of ginkgo biloba nuts and their nutritional value. Also, some more details in a discussion should be added. The work has a high cognitive value.

Detailed comments:

Title: Probably a sentence of “Insert the title of your article here” is not necessary?

Author response:

We thank reviewer for the careful work. It has been deleted.

Page 2, line 44: “amino acids” is repeated

Author response:

It has been deleted.

Page 3, line 2: there is a mistake in a word “degradation”

Author response:

It has been corrected.

Results and Discussion

Page 4 line 36: what was CP content reported by Deng [18]? It needs better information.

Author response:

It has been rewritten as follows:

The CP content of ginkgo nut, on average 19.7% of dry matter, was higher than that reported by Deng et al. [18], who found that the protein content was 11.6% in defatted ginkgo flour.

Page 4 lines 7-8: a discussion needs deeper explanation

Author response:

We think the reviewer wanted to refer to “Page 4 lines 37-8”.

The reason for the difference on protein content in ginkgo seed might be due to the differences in analytical procedure and data expression. In the study by Deng et al., protein content was obtained by an alkaline dissolving and acid precipitating method, determined by the Bradford method, and the data was reported on defatted ginkgo flour. In our study, the crude protein content was determined by a Perkin-Elmer 2400 automatic element analyzer with conversion factor of 6.25, and the protein content was reported on a dry weight basis.

Page 4, line 5: there is “Pre” but shall be “Pro”

There are errors in the spelling of the word "table":

Page 7, lines 2, 18

Page 8, line 2

Author response:

We are very ashamed of these stupid mistakes. All have been corrected.

Reviewer: 2

Comments to the Author(s)

This manuscript describes protein content and amino acids profile in ten cultivars of ginkgo (*Ginkgo biloba* L.) nut from China. The manuscript will be acceptable after minor revision.

Page 1, Summary and throughout the whole manuscript, including Table 2, all the analytical data should be given the three significant figures, e.g. 22.13 should be 22.1.

Author response:

We thank reviewer for reminding this detail, and have corrected.

Page 2, section 3.1. Samples and sample preparation

How many fruits and/or what amounts of fruits were sampled from how many plants of each cultivar should be given.

Author response:

Approximate eighty fruits were harvested randomly from three plants of each cultivar. It has been added in Materials and Methods

Page 3, section 4. Results and Discussion

The authors have not given any explanation/discussion as to why there is difference in the amino acid and protein contents of different cultivars of the plant? It will be useful to correlate the variations in amino acids with soil parameters (composition), climatic conditions, rain fall, temperature, pH, fertilizers used and so on.

Author response:

In this paper, the ten cultivars analyzed were cultivated in Pizhou Ginkgo biloba Seedling Base under the same agronomic conditions. The only difference is the cultivar, genetic background. Indeed, there are various factors causing the difference in the amino acid and protein contents. In the next study, we plan to compare the same cultivar from different planting regions, where the correlation of the variations in protein and amino acids of ginkgo nuts with soil parameters and climatic conditions will be discussed.